# Symmetry, Gauss-Newton, and Whitening in Neural Network Optimization

**Vedanth M. Nilabh**  **Robin Walters**
Khoury College of Computer Sciences
Northeastern University
{nilabh.v, r.walters}@northeastern.edu

## Abstract

Neural networks have architecture dependent parameter space symmetries where distinct parameters realize the same function. If an optimizer does not behave consistently under symmetry transformations, optimization dynamics depend on the arbitrary choice of representative along symmetry orbits. Recent work characterizes symmetries via the Jacobian of the evaluation map, decomposing parameter space into a *functional subspace* affecting outputs and a *fiber* subspace of symmetry directions. We show Gauss-Newton (GN) is symmetry-equivariant and that symmetry equivariance forces the pullback form $P = J^\top B J$ within a natural class of preconditioners. We then show that GN uniquely achieves optimal convergence conditioning within this class, while whitening achieves optimal whitening conditioning (isotropy of updates) but exhibits structured symmetry breaking due to the square root. These results clarify the relationship between Gauss-Newton, whitening and symmetries, which could be especially valuable for overparameterized networks.

## 1 Introduction

Parameter symmetries can be formalized concretely through the Jacobian of the evaluation map $J(\theta) = \nabla_\theta f(\theta, x)$, whose rank defines the *functional dimension* and whose row space and kernel space define the functional and fiber subspaces, the latter of which corresponds to symmetry directions (Grigsby & Lindsey, 2024).

Matrix whitening methods approximate isotropic conditioning by learning preconditioners $P \approx \mathbb{E}[gg^\top]^{-1/2}$, which improve convergence (Frans et al., 2025b), but do not explicitly respect the functional/fiber decomposition due to their approximations (Frans et al., 2025a). Natural gradient achieves invariance under smooth reparameterizations of the parameter space (Amari, 1998). While related, reparameterization invariance is not identical to symmetry equivariance, and estimating the Fisher information reliably in highly overparameterized networks remains challenging (Kristiadi et al., 2023) (van Oostrum et al., 2022). Recent work has positioned Gauss-Newton as an upper bound on performance for second-order optimizers (Abreu et al., 2025). Since matrix whitening approximates the related $G^{1/2}$, this raises the question: why would we observe a performance gap between these methods and full Gauss-Newton beyond approximation error and how do both relate to symmetry?

We ask: *which preconditioners respect neural parameter symmetries, what constraints does this impose and how does this relate to conditioning and convergence behavior?* Our contributions:

- We prove Gauss-Newton is exactly symmetry-equivariant (Theorem 1), and that this forms a broad class of natural symmetry equivariant preconditioners of the pullback form $P = J^\top B J$ (Theorem 2), which is implicitly shown in prior work (Martens, 2020).
- We prove that GN uniquely achieves optimal Newton-style convergence conditioning within this class, while $G^{1/2}$ achieves optimal whitening/isotropy via the congruence transform but exhibits structured symmetry breaking (Theorem 3 and 4).
- We clarify the relationship between whitening, GN, and Fisher, and the distinction between symmetry equivariance and reparameterization invariance.

## 2 PRELIMINARIES

Let $\theta \in \mathbb{R}^d$ denote parameters, $f(\theta, x) \in \mathbb{R}^k$ the model output, and $\ell(z, y)$ a loss twice-differentiable in its first argument. We consider losses $L(\theta) = \frac{1}{n} \sum_{i=1}^n \ell(f(\theta, x_i), y_i)$.

The **evaluation Jacobian** $J(\theta) := \nabla_\theta f(\theta, x) \in \mathbb{R}^{k \times d}$ induces a decomposition: the **functional subspace** $\mathcal{F}_\theta := \text{Im}(J^\top)$ contains directions affecting outputs, while the **fiber** $\mathcal{N}_\theta := \ker(J)$ contains symmetry directions. By the chain rule, $g(\theta) := \nabla_\theta L = J^\top \nabla_z \ell \in \mathcal{F}_\theta$.

The **Gauss-Newton matrix** is $G(\theta) := J^\top H_z J$, where $H_z := \nabla_z^2 \ell$, known as the loss hessian, is positive semidefinite for convex losses, such as Mean Squared Error (MSE) and Cross Entropy (CE). This is an example of a **pullback**: given a matrix $B$ on output space $\mathbb{R}^k$, the pullback through $J$ is $J^\top B J$, a matrix on parameter space. The pullback transfers geometry from output space (often thought of as a Riemannian manifold) to parameter space via the evaluation Jacobian.

**Parameter symmetries.** A *parameter symmetry* is an invertible map $\phi : \mathbb{R}^d \to \mathbb{R}^d$ satisfying $f(\phi(\theta), x) = f(\theta, x)$ for all $\theta, x$. We restrict our attention to smooth symmetries, which are diffeomorphisms that leave the function unchanged and let $T(\theta) := D\phi(\theta)$ denote its Jacobian. Differentiating both sides of the symmetry condition with respect to $\theta$ and applying the chain rule gives $J(\phi(\theta))T = J(\theta)$, so $J(\phi(\theta)) = J(\theta)T^{-1}$.

**Symmetry-equivariant preconditioners.** The gradient $g = J^\top \nabla_z \ell$ transforms as $g \mapsto T^{-\top} g$ under symmetry (since $J \mapsto JT^{-1}$). For the update $\Delta\theta = -P^{-1}g$ to transform as $\Delta\theta \mapsto T\Delta\theta$, we need $P$ to satisfy $P(\phi(\theta)) = T^{-\top} P(\theta)T^{-1}$, which is also a congruence transformation of $P$ by $T$. A preconditioner with this property is called *symmetry-equivariant*.

Why does this make updates equivariant? The gradient transforms as $g \mapsto T^{-\top}g$ (Theorem 1), so:

$$\Delta\theta' = -(P')^{-1}g' = -(T^{-\top}PT^{-1})^{-1}(T^{-\top}g) = -TP^{-1}g = T\Delta\theta.$$

**Why symmetry equivariance matters.** If $\theta$ and $\phi(\theta)$ realize the same function, a sensible optimizer should always give the same functional trajectory. Symmetry equivariance guarantees this. Without it, optimization dynamics depend on the arbitrary choice of representative within a symmetry orbit. In overparameterized networks where symmetry orbits are large, this can cause the optimizer to behave inconsistently across equivalent initializations or be sensitive to optimizer hyperparameter selection.

## 3 MAIN RESULTS

### 3.1 GAUSS-NEWTON IS EXACTLY SYMMETRY-EQUIVARIANT

**Theorem 1** (Exact symmetry equivariance of GN). *Under a parameter symmetry $\phi$ with Jacobian $T(\theta) = D\phi(\theta)$:*

$$g(\phi(\theta)) = T(\theta)^{-\top} g(\theta), \tag{1}$$
$$G(\phi(\theta)) = T(\theta)^{-\top} G(\theta)\, T(\theta)^{-1}. \tag{2}$$

*If $G(\theta)$ is invertible on $\mathcal{F}_\theta$, the GN update $\Delta\theta = -G^{-1}g$ satisfies*

$$\Delta\theta(\phi(\theta)) = T(\theta)\, \Delta\theta(\theta).$$

Note this result is implicit in Martens (2020) but is restated for parameter symmetries. The proof (Appendix A.1) follows from the chain rule and the symmetry condition.

**Why GN is natural.** The pullback structure $G = J^\top H_z J$ ensures $\ker(G) = \ker(J)$: GN has zero curvature exactly in fiber directions. The loss Hessian $H_z$ ensures it respects curvature in output space, making GN a valid Hessian approximation under small residuals. Thus GN is both symmetry equivariant and a principled curvature approximation.

## 3.2    Symmetry Equivariance Forces Pullback Structure

**Assumption 1** (Dependence on the evaluation map). *The preconditioner $P(\theta)$ depends on $\theta$ only through the evaluation map $f(\theta, \cdot)$ and its derivatives. Consequently, directions that do not change the function are ignored, i.e. $\ker J(\theta) \subseteq \ker P(\theta)$.*

**Theorem 2** (Pullback necessity). *If $P(\theta)$ is symmetry-equivariant and satisfies Assumption 1, then on $\mathcal{F}_\theta$ there exists a matrix $B_\theta$ depending only on the evaluation map such that*

$$P(\theta) = J(\theta)^\top B_\theta \, J(\theta).$$

Note this result is stated in terms of reparameterization here Martens (2020) but is restated for clarity of discussion.

## 3.3    Whitening: Structured Symmetry Breaking

Define

$$S(\theta) := T(\theta)^{-1} T(\theta)^{-\top} \succ 0,$$

which measures the distortion induced by symmetry.

Consider the matrix whitening update

$$\Delta_{\mathrm{w}}(\theta) := -G(\theta)^{-1/2} g(\theta),$$

where $G^{-1/2}$ denotes the principal (pseudo-)inverse square root on the functional subspace $\mathcal{F}_\theta$.

**Theorem 3** (Equivariance defect of matrix whitening). *Under a parameter symmetry $\phi$ with tangent map $T(\theta)$, the whitening update transforms as*

$$\Delta_{\mathrm{w}}(\phi(\theta)) = T(\theta) \, M(\theta) \, \Delta_{\mathrm{w}}(\theta),$$

*where the defect operator $M(\theta)$ is invertible, given by*

$$M(\theta) := S(\theta) \, G(\theta)^{1/2} \big( G(\theta)^{1/2} S(\theta) G(\theta)^{1/2} \big)^{-1/2}.$$

*In particular, whitening is exactly symmetry-equivariant iff $M(\theta) = I$, which holds when $T(\theta)$ is orthogonal, just as in vanilla gradient descent ($S = I$). See Appendix A.3 for proof.*

**Corollary 1** (Distortion suppression relative to gradient descent). *Let $\kappa(S)$ denote the condition number of the symmetry distortion operator $S$. Then the whitening defect satisfies*

$$\kappa(M) = \sqrt{\kappa(S)}.$$

*In contrast, gradient descent obeys*

$$\Delta_{\mathrm{gd}}(\phi(\theta)) = T(\theta) \, S(\theta) \, \Delta_{\mathrm{gd}}(\theta),$$

*so its symmetry distortion scales with $\kappa(S)$ directly.*

## 3.4    Optimal Conditioning Within the Pullback Class

Since symmetry-equivariant preconditioners must take the pullback form $P = J^\top B J$, we ask which choice of $B$ yields desirable conditioning.

Consider the quadratic model

$$L(\theta + \Delta) \approx L(\theta) + g^\top \Delta + \tfrac{1}{2}\Delta^\top G \Delta.$$

Two objectives arise:

**Newton conditioning.** Choosing $P = G$ gives $\Delta = -G^{-1}g$, the Newton step.

**Whitening conditioning.** Whitening requires isotropic updates, $P^{-T}\mathbb{E}[gg^\top]P^{-1} = I$. Since $\mathbb{E}[gg^\top] \approx G$, this gives $P = G^{1/2}$.

**Theorem 4** (Conditioning objectives). *Let $P(\theta) = J^\top B J$ and $G(\theta) = J^\top H_z J$. On $\mathcal{F}_\theta$:*

1. **Newton-style:** $P = G$ iff $B = H_z$.

2. **Whitening-style:** $P^{-T}GP^{-1} = I$ iff $P = G^{1/2}$ and there is no $B$ to write $P$ as a pullback

Thus Gauss–Newton ($B = H_z$) achieves Newton conditioning, while whitening produces $P = G^{1/2}$, which is not symmetry-equivariant generally. Practical whitening optimizers approximate $\mathbb{E}[gg^\top]^{-1/2}$, corresponding to approximating $G^{1/2}$ (Appendix A.6).

## 4 DISCUSSION

### 4.1 GN, FISHER, AND REPARAMETERIZATION

Natural gradient is defined using the Fisher information matrix

$$F = \mathbb{E}[\nabla \log p_\theta \, \nabla \log p_\theta^\top],$$

which requires the model to define a probability distribution. In practice it is often approximated by the empirical Fisher, which depends on labels rather than the model distribution and can differ substantially from the true Fisher (Kunstner et al., 2020).

Following Kristiadi et al. (2023), a *reparameterization* $\phi : \Theta \to \Psi$ is a diffeomorphism between parameter spaces under which natural gradient is invariant. A *symmetry* $T : \Theta \to \Theta$ instead acts within one parameter space while leaving the function unchanged ($f(T\theta) = f(\theta)$), and thus forms a special case of reparameterization.

For common losses such as cross-entropy, the Fisher and Gauss–Newton matrices coincide. Both respect smooth reparameterizations and therefore parameter symmetries. In practice, Gauss–Newton has proven easier to approximate in large networks, whereas natural-gradient methods often struggle with reliable Fisher estimation in highly overparameterized settings (Abreu et al., 2025; Kunstner et al., 2020; van Oostrum et al., 2022).

### 4.2 PRACTICAL CONSIDERATIONS

Practical whitening optimizers such as Shampoo and Muon use structured approximations (e.g., Kronecker factorizations and layerwise whitening) (Frans et al., 2025a). These generally violate the pullback structure $P = J^\top BJ$ and therefore break symmetry equivariance. In practice, J is typically singular in overparameterized networks, which can introduce additional sources of symmetry breaking, but GN has been shown to be close to optimal for relatively large transformers (Abreu et al., 2025). Our analysis suggests whitening methods implicitly trade symmetry equivariance for isotropic updates. This can improve robustness to noise but deviates from the Gauss–Newton geometry, which fully inverts curvature on the functional subspace.

## 5 CONCLUSION

Overparameterized networks contain large symmetry orbits: many parameter configurations represent the same function. A symmetry-equivariant optimizer therefore produces consistent functional trajectories across equivalent parameterizations.

Gauss–Newton satisfies this property because its pullback structure $G = J^\top H_z J$ yields $\ker(G) = \ker(J)$. More generally, within a natural class of functional preconditioners, symmetry equivariance forces the pullback form $P = J^\top BJ$. Within this class, Gauss–Newton and whitening correspond to two distinct desirable conditioning objectives: curvature inversion and isotropic updates.

Whitening achieves isotropy but breaks symmetry equivariance due to the matrix square root, introducing controlled distortion relative to gradient descent. These results clarify the geometric trade-offs underlying second-order and whitening-based optimizers in overparameterized networks.

## ACKNOWLEDGMENTS

We thank Elisenda Grigsby, Kathryn Lindsey, Purvik Patel, and Lucas Laird for helpful discussions.

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

## A    PROOFS

### A.1    PROOF OF THEOREM 1

Let $\phi$ be a symmetry with $f(\phi(\theta), x) = f(\theta, x)$ and $T = D\phi(\theta)$.

**Jacobian.** Differentiating $f(\phi(\theta), x) = f(\theta, x)$ with respect to $\theta$ gives $J(\phi(\theta))T = J(\theta)$, so $J(\phi(\theta)) = J(\theta)T^{-1}$.

**Gradient.** By the chain rule, $g = J^\top \nabla_z \ell$. Since outputs match, $\nabla_z \ell$ is unchanged while $J \mapsto JT^{-1}$:
$$g(\phi(\theta)) = (JT^{-1})^\top \nabla_z \ell = T^{-\top} J^\top \nabla_z \ell = T^{-\top} g(\theta).$$

**GN matrix.** Similarly $H_z$ is unchanged, so:
$$G(\phi(\theta)) = (JT^{-1})^\top H_z (JT^{-1}) = T^{-\top} G(\theta) T^{-1}.$$

**Update.** On $\mathcal{F}_\theta$ where $G$ is invertible:
$$\Delta\theta(\phi(\theta)) = -G(\phi(\theta))^{-1} g(\phi(\theta)) = -(T^{-\top} G T^{-1})^{-1}(T^{-\top} g) = -TG^{-1}g = T\Delta\theta(\theta).$$
$\square$

### A.2    PROOF OF THEOREM 2

We show that symmetry equivariance of the update $\Delta\theta = -P^{-1}g$, together with fiber-blindness and dependence only on the evaluation map, forces a pullback structure.

Since $\ker J \subseteq \ker P$, the preconditioner acts only on the functional subspace $\mathcal{F}_\theta = \mathrm{Im}(J^\top)$, as directions in $\ker J$ do not affect the function. On this subspace, the Jacobian $J$ is injective. Consequently any such preconditioner can be written in the form
$$P = J^\top B J$$
for some matrix $B$ acting on $\mathrm{Im}(J)$.

**Symmetry constrains $B$.**    If $P$ is equivariant, then under any symmetry $\phi$:

$$J(\phi(\theta))^\top B_{\phi(\theta)} J(\phi(\theta)) = T^{-\top} J^\top B_\theta J T^{-1}.$$

Using $J(\phi(\theta)) = J T^{-1}$ and simplifying: $J^\top B_{\phi(\theta)} J = J^\top B_\theta J$. Since $J$ is injective on $\mathcal{F}_\theta$, we have $B_{\phi(\theta)} = B_\theta$ on $\mathrm{Im}(J)$. Combined with Assumption 1, $B$ depends only on $f(\theta)$. $\quad\square$

### A.3   PROOF OF THEOREM 3

From Theorem 1,

$$g' = T^{-\top} g, \qquad G' = T^{-\top} G T^{-1}.$$

Define $B := G^{1/2} T^{-1}$. Then

$$G' = B^\top B.$$

Using $g = G^{1/2} G^{-1/2} g$, we write

$$g' = T^{-\top} g = B^\top (G^{-1/2} g).$$

Thus the whitening update transforms as

$$\Delta_{\mathrm{w}}(\phi(\theta)) = -(G')^{-1/2} g' = -(B^\top B)^{-1/2} B^\top (G^{-1/2} g).$$

Using the identity

$$(B^\top B)^{-1/2} B^\top = B^\top (B B^\top)^{-1/2},$$

we obtain

$$\Delta_{\mathrm{w}}(\phi(\theta)) = -B^\top (B B^\top)^{-1/2} (G^{-1/2} g).$$

Since

$$B B^\top = G^{1/2} T^{-1} T^{-\top} G^{1/2} = G^{1/2} S G^{1/2},$$

and

$$B^\top = T^{-\top} G^{1/2}$$

we obtain

$$\Delta_{\mathrm{w}}(\phi(\theta)) = -T\Big( S G^{1/2} (G^{1/2} S G^{1/2})^{-1/2} \Big)(G^{-1/2} g).$$

Defining

$$M = S G^{1/2} (G^{1/2} S G^{1/2})^{-1/2},$$

gives

$$\Delta_{\mathrm{w}}(\phi(\theta)) = T M \Delta_{\mathrm{w}}(\theta).$$

$\quad\square$

### A.4   PROOF OF COROLLARY 1

Write the polar decomposition of $G^{1/2} S^{1/2} = U D$ where $U$ is orthogonal and $D \succ 0$. Then

$$G^{1/2} S G^{1/2} = (G^{1/2} S^{1/2})(G^{1/2} S^{1/2})^\top = U D^2 U^\top,$$

so $(G^{1/2} S G^{1/2})^{-1/2} = U D^{-1} U^\top$. Substituting into $M = S G^{1/2} (G^{1/2} S G^{1/2})^{-1/2}$:

$$M = S^{1/2}(S^{1/2} G^{1/2}) U D^{-1} U^\top = S^{1/2}(G^{1/2} S^{1/2})^\top U D^{-1} U^\top = S^{1/2} D U^\top U D^{-1} U^\top = S^{1/2} U^\top.$$

Since $U$ is orthogonal, the singular values of $M$ equal those of $S^{1/2}$, so

$$\kappa(M) = \frac{\lambda_{\max}(S)^{1/2}}{\lambda_{\min}(S)^{1/2}} = \sqrt{\kappa(S)}.$$

The gradient descent defect is $S$ directly, so $\kappa(S) = (\sqrt{\kappa(S)})^2$, confirming whitening halves the distortion exponent. $\quad\square$

## A.5   Proof of Theorem 4

Let $P = J^\top B J$ and $G = J^\top H_z J$. We work on $\mathcal{F}_\theta$ where $J$ is injective.

**Newton ($P = G$).**   We need $J^\top B J = J^\top H_z J$. Since $J$ is injective on $\mathcal{F}_\theta$, this holds iff $B = H_z$.

**Whitening ($P = G^{1/2}$).**   We require $P^{-T} G P^{-1} = I$, equivalently $P^2 = G$ (since G is symmetric). Within the pullback class $P = J^\top B J$, this would require

$$J^\top B J J^\top B J = J^\top H_z J,$$

which cannot be satisfied while preserving symmetry-equivariance, since matrix square roots do not commute with congruence transformations in general. Thus no symmetry-equivariant pullback preconditioner achieves exact whitening.

## A.6   Relating Whitening to $G^{\frac{1}{2}}$

**The whitening metric.**   With $g = J^\top \nabla_z \ell$, the gradient covariance factors as:

$$\mathbb{E}[gg^\top] = J^\top \mathbb{E}[\nabla_z \ell \, \nabla_z \ell^\top] J.$$

This is a pullback through $J$, so whitening inherits the pullback structure.

**Cross-entropy loss.**   For cross-entropy loss, $\nabla_z \ell = p - y$ where $p$ is the predicted distribution. Under the model distribution $y \sim p$:

$$\mathbb{E}_{y \sim p}[\nabla_z \ell \, \nabla_z \ell^\top] = \mathrm{diag}(p) - pp^\top = H_z.$$

So $\mathbb{E}[gg^\top] = G$, and ideal whitening gives $G^{1/2}$. This also equals the Fisher information matrix under the model distribution, so Gauss-Newton and Fisher coincide for CE (Kunstner et al., 2020).

**MSE loss.**   For MSE, $\nabla_z \ell = f(\theta, x) - y =: r$ (the residual) and $H_z = I$. Then $\mathbb{E}[gg^\top] = J^\top \mathbb{E}[rr^\top] J$, which equals $G = J^\top J$ only if residuals are isotropic. So whitening approximates but does not equal $G^{1/2}$.

Only $G$ (Newton-optimal) but not $G^{1/2}$ (whitening-optimal) is symmetry-equivariant in the idealized case.

