# OpenReview forum: "Symmetry, Gauss-Newton, and Whitening in Neural Network Optimization"
_ICLR.cc/2026/Workshop/GRaM — ICLR 2026 Workshop GRaM Poster_

### Official Review · Reviewer_9Vys · 2026-02-21

**Rating:** 4
**Confidence:** 4

**Review:**

Strengths:
Looking at parameter symmetries and how they relate to optimizers is extremely important. The math is correct. The results on whitening are of particular interest and are the strongest part of the paper.

Weaknesses:
The notational choices make this paper quite hard to read, the variables changing names throughout the paper is difficult to keep track of. There are also some minor technical notes that should be added for complete rigor, I've put the specifics below.

The main theorem about invariance of Gauss-Newton is known already, in the papers that are cited in this paper.

Comments:
068 - It should be mentioned that the parameter symmetries under consideration are those which are diffeomorphisms, not just smooth + bijective.
105 - Why does equality of kernels hold here? I believe one can only ensure that $\ker(J) \subseteq \ker(G)$, which is really enough for the discussion following.
122 - Possibly I'm misunderstanding Assumption 1, but it seems that Assumption 2 follows from it in any reasonable definition.
137 - The Gauss-Newton approximation was called $G$ previously, here it switches to being called $H$ which is slightly confusing.
152,155, etc... - Switches back to being $G$? Check this carefully through the paper.

**Pmlr Suitability:**

Yes

---

### Official Review · Reviewer_oWmv · 2026-02-24
**Review of Characterizing Symmetry Equivariant Preconditioners in Neural Network Optimization**

**Rating:** 6
**Confidence:** 3

**Review:**

# Summary
This paper explores the use of optimizers which are equivariant to parameter symmetries within a network. They show a number of results, including equivariance of Gauss-Newton preconditioning under parameter symmetries, the form of any preconditioner given natural assumptions, forms of preconditioners for different objectives, and equivariance of powers of Gauss-Newton matrix.

# Strengths
Understanding the role of parameter symmetries for optimization is very important. The results are presented in a relatively logical manner.

# Weaknesses
The results seem to be rather straightforward and modest. Some notation is not fully defined, in particular in Theorem 4.

# Conclusion
The topic is interesting though the results presented in the paper are relatively simple. I lean towards acceptance for a workshop tiny paper.

**Pmlr Suitability:**

NA

---

### Official Review · Reviewer_fueh · 2026-02-24
**Note on symmetry equivariant preconditions**

**Rating:** 6
**Confidence:** 2

**Review:**

This paper is about when preconditioners respect neural parameter symmetries. It shows Gauss-Newton is symmetry-equivariant, argues symmetry equivariance forces a pullback form, and contrasts Newton-style with whitening-style conditioning targets.

Strengths:
- The connection between the functional/fiber decomposition and optimizer design is well-framed.
- The distinction between symmetry equivariance and reparameterization invariance is a useful clarification.
- The whitening discussion helps clarify relationships between GN/Fisher/whitening metrics.

Weakness:
- I found the presentation somewhat dense for a tiny paper. A simple toy example of a common symmetry (permutation/scaling) and clearer notation in later theorems would improve readability.
- Practical implications are largely speculative without experiments.

I think it would be of relevance to GRaM.

**Pmlr Suitability:**

NA

---

### Meta-Review · Area_Chair_4qd2 · 2026-02-25

**Decision:**

Accept

**Metareview:**

Reviewers agree that the topic of the paper (the relation of parameter symmetries to optimisers) and the results on whitening are interesting and therefore I recommend acceptance. However, one reviewer points out that Theorem 1 in the paper is already known in the literature and should therefore be cited.

**Relevance To Proceedings:**

Tiny paper — does not apply

**Relevance To Workshop:**

Yes — suitable for GRaM

---

### Decision · Program_Chairs · 2026-03-02

Accept (Poster)